# PAXgene Fixation for Pancreatic Cancer: Implications for Molecular and Surgical Pathology

**DOI:** 10.3390/jcm11144241

**Published:** 2022-07-21

**Authors:** Ryan DeCoste, Yutaka Amemiya, Sarah Nersesian, Lauren Westhaver, Stacey N. Lee, Michael D. Carter, Heidi L. Sapp, Ashley E. Stueck, Thomas Arnason, Jeanette Boudreau, Arun Seth, Weei-Yuarn Huang

**Affiliations:** 1Department of Pathology & Laboratory Medicine, QEII Health Sciences Centre, Nova Scotia Health Authority (Central Zone), Halifax, NS B3H 1V8, Canada; ryan.decoste@nshealth.ca (R.D.); michaeld.carter@nshealth.ca (M.D.C.); heidi.sapp@nshealth.ca (H.L.S.); ashleye.stueck@nshealth.ca (A.E.S.); thomas.arnason@nshealth.ca (T.A.); 2Department of Pathology, Dalhousie University, Halifax, NS B3H 1V8, Canada; lauren.westhaver@dal.ca (L.W.); jeanette.boudreau@dal.ca (J.B.); 3Sunnybrook Research Institute Genomics Core Facility, Sunnybrook Health Sciences Centre, Toronto, ON M4N 3M5, Canada; yamemiya@sri.utoronto.ca (Y.A.); arun.seth@sunnybrook.ca (A.S.); 4Department of Microbiology & Immunology, Dalhousie University, Halifax, NS B3H 4R2, Canada; s.nersesian@dal.ca (S.N.); stacey.lee@dal.ca (S.N.L.); 5Department of Laboratory Medicine & Molecular Diagnostics, Sunnybrook Health Sciences Centre, Toronto, ON M4N 3M5, Canada; 6Department of Laboratory Medicine and Pathobiology, University of Toronto, Toronto, ON M5S 1A8, Canada

**Keywords:** PAXgene, NGS, pancreatic cancer, molecular diagnostics, surgical pathology

## Abstract

Genomic profiling of pancreatic cancer using small core biopsies has taken an increasingly prominent role in precision medicine. However, if not appropriately preserved, nucleic acids (NA) from pancreatic tissues are known to be susceptible to degradation due to high intrinsic levels of nucleases. PAXgene fixation (PreAnalytix, Switzerland) represents a novel formalin-free tissue preservation method. We sought to compare the NA and histomorphological preservation of pancreatic cancer tissues preserved with PAXgene-fixed paraffin-embedding (PFPE) and formalin-fixed paraffin-embedding (FFPE). Tissues from 19 patients were obtained prospectively from pancreaticoduodenectomy specimens and evaluated by four gastrointestinal pathologists. The extracted NA were quantified by Nanodrop and Qubit and assessed for quality by qPCR, targeted next-generation sequencing (NGS) assay, and RNA-sequencing. Our results demonstrated that, when assessed blindly for morphological quality, the four pathologists deemed the PFPE slides adequate for diagnostic purposes. PFPE tissues enable greater yields of less fragmented and more amplifiable DNA. PFPE tissues demonstrated significantly improved quality control (QC) metrics in a targeted NGS assay including Median Absolute Pair-wise Difference (MAPD) scores. Our results support the use of PAXgene fixative for the processing of specimens from pancreatic cancers with the potential benefits of improved yields for more amplifiable DNA in low-yield biopsy specimens and its ideal use for amplicon-based NGS assays.

## 1. Introduction

Pancreatic ductal adenocarcinoma (PDAC) remains one of the most lethal human malignancies. While recent work in academic research has revealed some of the underlying molecular abnormalities in PDAC, there is no specific framework with which to guide patient management based on molecular findings [1]. Furthermore, clinical efforts may be limited by the feasibility of applying molecular biomarker testing on a case-by-case basis, including issues of tissue preservation for various methods of analysis.

Formalin-fixed paraffin-embedded (FFPE) specimen processing is currently the standard in surgical pathology labs as it is known to provide excellent histomorphological preservation, which is critical to confidently diagnosing PDAC. Immunohistochemistry on FFPE may be limited by protein cross-linking; however, these techniques are optimized for FFPE in pathology laboratories to provide quality, reproducible immunohistochemical analysis. Unfortunately, DNA and RNA extracted from FFPE samples is suboptimal in yield and quality. Fresh frozen (FF) tissue is known to yield excellent molecular preservation. However, tissue freezing impairs histomorphological analysis and may not be practical when applied to large tumour resection specimens such as pancreaticoduodenectomy (Whipple) resections for PDAC [2,3].

Although nucleic acid quality from FFPE samples of many cancer sites is acceptable to perform next-generation sequencing (NGS) in routine clinical practice, the nucleic acid quality of FFPE PDAC samples has not been thoroughly examined. Considering the microenvironment present in the pancreas, with numerous degradative enzymes including nucleases, it is crucial to address if DNA and RNA from pancreatic FFPE tissues remain stable for cancer biomarker testing. In addition, it is important to consider whether there may be a better fixative than Formalin to yield enough DNA/RNA from fine needle biopsies without compromising histomorphology.

A recently developed methanol and acetic acid-based fixative, PAXgene (PAXgene Tissue Fix Container, Preanalytix GmbH, Hombrechtikon, Switzerland), works via a non-crosslinking mechanism. Studies involving multiple types of cancer have demonstrated that PAXgene-fixed, paraffin-embedded (PFPE) tissue yields adequate histomorphology in the majority of cases (i.e., comparable to FFPE, far better than frozen tissue) [4,5,6,7,8,9,10]. These studies also suggested PFPE is associated with improved DNA and RNA yield and quality compared to FFPE [6,7,8,9,10,11]. In addition, three cancer morphology ring studies under the SPIDIA consortium (standardization and improvement of generic pre-analytical tools and procedures for in vitro diagnostics) focusing on colon, breast and prostate have been executed and demonstrated promising advantages using the PAXgene fixative in routine pathology in comparison to formalin fixation [4,8,12]. However, there remain limited numbers of studies addressing the applicability of nucleic acids extracted from PFPE tissues on targeted next-generation sequencing (NGS) panels that have been fully optimized for FFPE samples. To date, no study has reported a direct comparison of FFPE and PFPE molecular integrity and histomorphological quality, in a blinded manner, in a substantial number of PDAC cases. In contrast to the above three cancer types which tend to be detected at a resectable stage, more than 50% of patients with pancreatic cancer are known to present at an advanced unresectable stage and the final pathology diagnosis, including biomarker studies, often relies on a small fine-needle biopsy specimen [1]. We believe that it is imperative to evaluate if PAXgene fixation could also demonstrate advantages over formalin fixation in morphology and genomic profiling via a targeted NGS panel optimized for FFPE clinical samples, using a small amount of pancreatic tumor tissues known to have enriched intrinsic nuclease activity.

## 2. Methods

This study was approved by the Nova Scotia Health Authority (NSHA) institutional research ethics board in accordance with regional tissue banking practices (#1023554; approval date: 19 June 2018). All patients consented to participating in this study. Specimens were collected in the period from September 2018 to October 2019. Inclusion criteria included any case in which acquisition of a pancreaticoduodenectomy specimen was accompanied by consent for tumor banking, and in which there was a suspected or confirmed diagnosis of PDAC. Exclusion criteria included a lack of consent for tumor banking, an alternative suspected or confirmed diagnosis, tumor size too small to safely sample for tumor banking or the current study protocol without sacrificing diagnostic tissue, and cases in which a tumor could not confidently be identified grossly.

### 2.1. Collection and Fixation of Tumor Tissue

Fresh pancreaticoduodenectomy specimens without fixation were received in the anatomical pathology laboratory. Following gross specimen assessment, inking, and sectioning (axial technique) as per institutional protocols, tissue acquisition for principal tissue banking was performed. Additional samples were obtained for PAXgene fixation and comparison to formalin fixation via punch biopsy (5 mm punch). Each core was bisected—one half for PAXgene fixation and the other half for formalin fixation. All samples were approximately equal in size, were within the maximum tissue dimensions (15 × 15 × 4 mm) recommended for utilizing PAXgene fixation, and did not include adipose tissue. Formalin-fixed samples were placed in a tissue cassette, fixed and processed along with the remainder of the case using standard institutional protocols, such that a representative section would be received by the pathologist in charge of diagnosis. Following fixation of PAXgene samples for 24 h, they were immersed in PAXgene Tissue Stabilizer and stored refrigerated (refrigerated at ~4 °C) until processing. Refrigerated storage times in the Tissue Stabilizer ranged from 1 to 4 weeks. PAXgene fixation, stabilization, and storage of PAXgene samples were performed in accordance with manufacturer instructions.

### 2.2. Tissue Processing

Tissue processing was performed with an automated program performed on a Tissue-Tek VIP instrument (Sakura Finetek USA, Inc., Torrance, CA, USA) utilizing an aqueous-free pathway for PAXgene-fixed samples, resulting in the creation of PFPE and FFPE tissue blocks in a standard fashion. A representative 5-micrometer section was obtained from each and stained with Hematoxylin and Eosin on a Ventana HE 600 automated staining system (Roche Diagnostics, Basel, Switzerland) according to a standard laboratory protocol.

### 2.3. Morphological Analysis

De-identified H&E-stained slides blinded to fixation methods were reviewed by four pathologists with subspeciality training and/or a significant practice in gastrointestinal pathology, who completed a brief survey probing the quality of each individual slide analyzed, and whether the pathologist could identify the method of fixation (Table 1). In addition, room was provided for additional comments at the end of the survey. Each evaluator assessed a total of 38 histology slides (19 blinded PFPE samples, 19 blinded FFPE samples).

### 2.4. Nucleic Acid Sample Preparation and Quantification

A portion of each paraffin-embedded tissue sample was obtained via 3 mm punch biopsy of the paraffin block. From each punched core of tissue, a 1 mm thick disc was cut for nucleic acid extraction. Nucleic acid was extracted from PFPE samples using the PAXgene Tissue DNA and RNA/miRNA kits, according to the manufacturer’s instructions. Nucleic acid was extracted from FFPE samples using Qiagen AllPrep FFPE/Mini kits (Hilden, Germany), according to the manufacturer’s instructions. The extracted genomic DNA (gDNA) and total RNA were quantified by Nanodrop and Qubit 3 Fluorometer using the Qubit dsDNA HS and RNA HS assay kits, respectively (Thermo Fisher Scientific Inc., Waltham, MA, USA). The amount of amplifiable gDNA was assessed by two qPCR assays through amplifying a 100 bp genomic region of *YWHAZ* (14-3-3 protein zeta/delta) gene and 104 bp genomic regions of Glucuronidase beta gene (*GUSB)*. Total RNA was assessed by amplifying a 96 bp coding region of *GUSB*. The qPCR for *GUSB* was carried out in a 10 μL reaction with 1 × TaqMan Fast Advance Mix, 1 × Human *GUSB* assay mix (Thermo Fisher Scientific Inc.) and 2 μL of extracted nuclei acid by using the StepOnePlus Real-Time PCR System (Thermo Fisher Scientific Inc., Waltham, MA, USA). Standard curves for gDNA and total RNA quantification with six-point serial dilution from 50 ng to 16 pg were prepared using TaqMan Control Genomic DNA Human (Cat#:4444434, Thermo Fisher Scientific Inc., Waltham, MA, USA) and HL-60 Total RNA (Cat#:AM7836, Thermo Fisher Scientific Inc., Waltham, MA, USA), respectively. For the 2nd qPCR assay, universal SYBR Green Supermix was used in combination with 50 ng gDNA and primers of *YWHAZ* gene (F: ACTTTTGGTACATTGTGGCTTCAA; R: CCGCCAGGACAAACCAGTAT). All samples were run in duplicate. Gel electrophoresis was used to verified primer specificity and amplicon size. Electrophoresis was performed using 1.5% agarose at 120V with RedSafe DNA dye and a100 bp ladder (FroggaBio) and imaged using the ChemiDoc Gel Imaging System (Biorad, Hercules, CA, USA). 

### 2.5. Targeted NGS Assay and Bioinformatic Analysis

The targeted NGS assay was performed on the Ion S5XL next-generation sequencing system with the Oncomine Comprehensive Assay v3 (OCAv3, Thermo Fisher Scientific Inc., Waltham, MA, USA). The OCAv3 assays includes full exon coverage of 48 genes, hotspot mutation detection of 87 genes and CNV detection of 43 genes. The amplicon library was constructed from 20 ng of gDNA by the Ion Ampliseq Library Plus Kit. Barcoded libraries were quantified using the Ion Library TaqMan Quantitation Kit (ThermoFisher Scientific Inc., Waltham, MA, USA) and diluted to a final concentration of 75 pM. The sequencing template preparation was performed using Ion Chef with Ion 540 Chef Kits. Sequencing was performed for 500 flows on an Ion S5XL Sequencer with an Ion 540 chip. The Ion Torrent platform-specific pipeline software, Torrent Suite version 5.16.0 (Thermo Fisher Scientific Inc., Waltham, MA, USA), was used to separate barcoded reads and to filter and remove polyclonal and low-quality reads. For the OCAv3 DNA sequencing, the overall quality of sequence data was assessed by number of mapped reads, % of on-target reads, mean sequencing depth, mean sequencing length and uniformity using the coverage analysis plug-in v5.16.0.4. BAM format files were generated from the sequencing results and then exported to the Ion Reporter Server (Thermo Fisher Scientific Inc., Waltham, MA, USA). The bioinformatics analysis of the sequence data was performed with Ion Torrent platform-specific bioinformatics software, Ion Reporter version 5.16 (Thermo Fisher Scientific Inc., Waltham, MA, USA). 

### 2.6. Transcriptome Sequencing

The transcriptome sequencing was performed on the Ion S5XL Next Generation Sequencing system with the Ion AmpliSeq Transcriptome Human Gene Expression Assay (ThermoFisher Scientifics Inc., Waltham, MA, USA). This assay covers 20,802 human RefSeq genes (>95% of UCSC ref Gene) with a single amplicon designed per gene target. The gDNA in the RNA sample was digested by EZDNase and the cDNA was synthesized from 20 ng of total RNA using SuperScript IV VILO Master Mix with the ezDNase Enzyme kit (Thermo Fisher Scientifics Inc., Waltham, MA, USA). The cDNA libraries were constructed by the Ion Ampliseq Library Kit 2.0 (Thermo Fisher Scientifics Inc., Waltham, MA, USA). Barcoded libraries were quantified using the Ion Library TaqMan Quantitation Kit (ThermoFisher Scientifics Inc., Waltham, MA, USA) and diluted to a final concentration of 80 pM. The sequencing template preparation was carried out using Ion Chef with Ion 540 Chef Kits (Thermo Fisher Scientifics Inc., Waltham, MA, USA). Sequencing was performed for 500 flows on an Ion S5XL Sequencer with an Ion 540 chip (Thermo Fisher Scientifics Inc., Waltham, MA, USA). Ion Torrent platform-specific plugin, ampliseqRNA version 5.16 (Thermo Fisher Scientifics Inc., Waltham, MA, USA) was used for the alignment of the raw sequencing reads and quantitation of normalized gene expression level (reads-per-million: RPM).

### 2.7. Statistical Analysis

Data were analyzed using Student’s T-test for continuous variables and Fisher’s Exact test for categorical variables, employing a cut-off of *p* ≤ 0.05 to indicate statistical significance.

## 3. Results

Paired samples were obtained from 19 cases for analysis, yielding 38 total specimens (19 PFPE, 19 FFPE). The cases included 13 cases of pancreatic ductal adenocarcinoma (68.4%), 4 ampullary adenocarcinomas (21.1%), and 2 intraductal papillary mucinous neoplasms (10.5%).

### 3.1. Tissue Morphology

Based on the analysis of answers to survey questions listed in Table 1, all observers deemed all H&E-stained sections included in the study to be adequate for diagnostic purposes in a morphology-based surgical pathology practice. Cumulatively, observers correctly identified a slide as being derived from FFPE or PFPE 53.9% of the time (82/152 total observations). Slides of PFPE tissue were correctly identified 55.3% of the time (42/76 total observations), and slides from FFPE tissue were correctly identified 52.6% of the time (40/76 total observations) (*p* = 0.87). Narrative free-text comments revealed that some observers noted a pale appearance with reduced contrast in a small subset of slides, which they believed to represent PFPE tissue. However, there was a lack of interobserver variability with inconsistent labelling of particular slides as pale. Upon further analysis, 4/19 (21.1%; 4 total observations) PFPE slides were described as pale and/or having poor contrast, while 6/19 (31.6%, 8 total observations) FFPE slides were described as such (*p* = 0.71). Examples of tissue morphology in FFPE and PFPE samples are displayed in Figure 1. Taken together, these results suggest that the morphological quality of PFPE with H&E staining is comparable to that of FFPE and is sufficient to make pathological diagnosis of common pancreatic tumors.

### 3.2. Nucleic Acid Quantification and Quality Assessment

Mean nucleic acid concentration measured by Nanodrop, Qubit assay and mean amplifiable nucleic acid concentration quantified by GUSB-qPCR assay are displayed in Table 2. Although there was no statistically significant difference in the yield of nucleic acid measured by either Nanodrop or Qubit assay, an almost 10-fold increase in the amount of amplifiable DNA using *GUSB* qPCR assay was observed in the PFPE tissue (*p* = 0.003). A second qPCR assay on the *YWHAZ* gene further supported the presence of more amplifiable gDNA from PFPE samples. As shown in Figure 2A,C, gel electrophoresis demonstrated the correct 100 bp of amplicon in both PFPE and FFPE samples. In the melting curve plots (Figure 2B,D), input of 20 ng of genomic DNA resulted in the mean Cq value of PFPE and FFPE tissues 20.24 and 22.73, respectively (*p* = 0.0001). For RNA analysis, there was a trend toward more amplifiable RNA in PFPE samples (7.9 vs. 5.5 ng/μL; *p* = 0.3), although this finding does not reach statistical significance indicating the presence of variations of yield of amplifiable RNA in individual samples.

### 3.3. Targeted NGS-DNA Assay

We next assessed the performance of targeted NGS assay using PFPE samples. Eight pairs of pancreatic ductal adenocarcinoma (eight FFPE and eight PFPE) with various tumor cellularity (10–50%) were tested using the OCAv3 DNA panel according to the protocol recommended by the manufacturer, in which DNA inputs were based on quantification by the Qubit assay. With 20 ng input of DNA, we noted that the yields after library preparations were significantly higher in PFPE samples than those of FFPE samples. The mean concentrations of PFPE and FFPE were 12,613 pM and 3053 pM, respectively (*p* = 0.0003), which resulted in the requirement of additional dilution of library preparations of PFPE samples before sequencing.

Comparing the QC metrics, PFPE samples exhibited significantly better performance in several parameters including read length, uniformity, percentage of targeted bases at 500×, and MAPD (Median of the Absolute values of all Pairwise Differences) metric. However, performance in total mapped reads and mean depth of coverage were similar between PFPE and FFPE samples (Table 3). Driver mutations of *KRAS* and *TP53* were detected in seven of eight pairs of samples. One of the paired samples also harbored two pathogenic *BRCA2* variants. Comparing FFPE to PFPE samples, there was no significant difference in the mean depth of coverage on these variants (1765 vs. 1993, *p* = 0.18).

### 3.4. RNA Sequencing

We next assessed the performance of four pairs of samples with similar tumor cellularity using the Ampliseq Transcriptome assay, which examines the RNA expression profile of more than 20,000 genes. The essential QC metrics demonstrated that the total mapped reads were similar between PFPE and FFPE tissues (9,162,356 vs. 9,078,672; *p* = 0.84) and improved mean read length in PFPE tissues compared to FFPE tissues (115 bp vs. 110 bp, *p* = 0.05). Based on library quantification by qPCR, the yield after library preparations was significantly higher in PFPE tissues compared to FFPE tissues (5733 pM vs. 2682 pM; *p* = 0.02). However, there remains variations in normalized reads for the individual genes across different samples without evidence of consistent higher expression of one fixation method over the other.

## 4. Discussion

In our study, all H&E-stained histology slides generated from PFPE tissue were assessed as being morphologically adequate for establishing a diagnosis of PDAC, ampullary adenocarcinoma, or intraductal papillary mucinous neoplasm in a pancreatic surgical pathology. PAXgene-fixed samples were able to be correctly distinguished from FFPE tissue in just over half of cases analyzed, a result only slightly better than chance alone would predict. Some observers noted a pale appearance with slightly reduced contrast that they believed may have derived from PFPE tissue. Ultimately, however, these comments were in the minority (e.g., described in 4 of 19 PFPE slides), and there was no significant difference in the proportion of slides that received such comments between the FFPE and PFPE samples. In fact, slightly more FFPE cases were described as pale in appearance. As a result, it seems likely that these observations derive from day-to-day variability in the performance of automated staining machines utilized in the laboratory, and if any real difference were to exist between fixatives, staining procedures would likely be amenable to optimization for individual fixation protocols. It should also be noted that interpretation of H&E staining quality is a subjective measure, with personal preferences among different pathologists. As such, all PFPE slides having been assessed as adequate for diagnostic purposes is likely a more meaningful metric. Our results are similar to those of other studies that have observed adequate morphological preservation with PAXgene fixation across variable tissue types [4,5,6,8]. Some studies have even shown a trend toward increased contrast in PAXgene-fixed samples [4,5]. 

A key finding in this study is that there are more amplifiable gDNA extracted from PFPE samples, although there is no significant difference in yields of overall gDNA between PFPE and FFPE samples quantified by either Nanodrop or the Qubit assay. The presence of more amplifiable gDNA and better mean read length in subsequent NGS testing indicates better DNA quality extracted from PFPE samples than that of FFPE samples. Not unexpectedly, compared to FFPE samples, PFPE samples based on the concentrations measured by the Qubit assay exhibited much higher yields (near 4-fold) after library preparations; as such, for NGS testing on PFPE samples, it would be more appropriate to apply the amounts for testing based on amplifiable gDNA measured by the qPCR assay, but not by the gold standard Qubit assay.

We also assessed the performance of PFPE samples using OCAv3, a targeted NGS panel that has been fully optimized for testing FFPE samples. Our results demonstrated that a targeted NGS assay is also applicable to PFPE samples with similar performances in certain essential QC metrics such as total mapped reads and mean depth of coverage. However, PFPE samples outperform FFPE samples in MAPD metrics, mean read length, percentage of targeted bases at 500× coverage and coverage uniformity. The MAPD metric is a measure of read coverage noise detected across all amplicons in a panel. Significantly improved MAPD metrics were noted in PFPE samples, suggesting higher coverage uniformity across all amplicons. MAPD metrics have been used widely by Affymetrics to assess the quality of CNV (copy number variation) calls [13]. A better MAPD metric (i.e., a lower MAPD score) is more ideal for CNV calls, as low coverage uniformity can result in missed or erroneous CNV calls.

To our knowledge, this is the first study using an Ion Torrent platform to assess the NGS performance of PFPE tissues. An earlier study by Högnäs et al. reported similar read counts between FFPE and PFPE prostatic tissues using a 36-gene targeted Miseq DNA sequencing assay [8]. However, other QC metrics were not evaluated in that study. Southwood et al., applying the Qiagen sequencing platform on PFPE lung tissues, reported superior NGS performance, both in terms of QC metrics and for variant calling with higher unique molecular identifier (UMI) reads and deeper mean depth of coverage than those of FFPE tissues [9]. We did not observe PFPE tissues exhibiting more mapped reads nor deeper mean depth of coverage using the Ion Torrent platform. The discrepancy in NGS performance could be attributed to differences in chemistry or design of amplicons. However, significantly, to our knowledge, we are the first to describe that PFPE tissues are more suitable for CNV calls than FFPE tissues. Given that accurate CNV calls of *BRCA1* and *BRCA2* are crucial to determine if certain variants are truly driver mutations in PDCA [1], it would be interesting to test if improved MAPD metrics of PFPE would result in accurate calling of exon loss or gain of *BRCA* genes in tumor samples using amplicon-based targeted NGS assays.

Comparing the results of quantity and quality of NA, we noted discrepancies in yields of NA among these studies. Although ours and Högnäs et al. concluded similar yields of gDNA between PFPE and FFPE tissue [8], Southwood et al. reported higher yield of DNA from PFPE tissues [9]. We did not observe consistently improved yield of RNA across different samples, while Högnäs et al. reported significantly improved RNA picogram yield per nucleus as compared with FFPE tissue [8]. Other studies assessing RNA quality from various tumor sites reported significantly improved RNA quality in PFPE tissues [5,6,7]. Our study on RNA sequencing also demonstrated a trend toward a better quality of RNA but noted some variations among PFPE tissues, suggesting that RNA quality from PFPE pancreatic tissues was not as stable as the other tissue types, which is likely attributable to the presence of abundant nuclease in the pancreatic tissue. Finally, there remains one consistent finding across different studies—better quality of gDNA supported by the presence of less fragmented DNA from PFPE tissues.

A formal cost–benefit analysis is beyond the scope of this study. However, we estimate that the cost for tissue fixation and routine processing with creation of an H&E-stained histology slide for PAXgene is CAD $21.97, vs. CAD $6.25 for formalin. The cost of PAXgene could conceivably be reduced if its use was scaled up due to larger purchasing power, but in the context of the current estimated costs, it is unlikely to replace formalin for routine surgical pathology specimens. Nevertheless, we argue that it could be employed on a selective basis with positive results, especially tailored to small biopsy specimens that have a high likelihood of requiring ancillary molecular diagnostic testing.

In a recent study, Groelz et al. demonstrated that PFPE tissues perform superiorly compared to FFPE tissues, regardless of storage time and temperature in both human and rat tissues [11]. We noted that pancreatic tissues were not included in that study. Although, our study demonstrated better preserved DNA from PFPE pancreatic tumors in a short-term storage condition (i.e., room temperature and less than 12 months old). It remains unknown if PAXgene fixation could also provide advantages over formalin-fixation on pancreatic tumors in a long-term storage condition, considering that formalin-fixation resulting in cross-linkage of NAs may provide better protection from intrinsic nucleases. Nevertheless, formalin fixation has its own drawback, as significant sequencing artifacts have been reported in achieved FFPE tissues attributed to formalin over-fixation [14]. Additional experiments on long-term storage conditions of pancreatic tumors and their effects on NGS analysis will clarify this matter.

One weakness of our study is that histochemical and immunohistochemical staining techniques were not evaluated. Prior studies have shown variable performance of immunohistochemical techniques on PFPE tissue, with some documented good performance with minor alteration of conventional laboratory protocols [4,5,8], and others demonstrating a need for further protocol optimization due to lower staining intensity [6]. Ultimately, it seems likely that the variability in such results may be due in part to the individual characteristics of different antibodies and clones thereof, and further study is necessary for PDCA, although pathological diagnosis of PDCA rarely requires immunohistochemistry.

Additional limitations of our study include a relatively small sample size and its focus on one diagnosis (PDAC). As such, the findings of our study alone are not necessarily generalizable to other tumor types (e.g., pancreatic neuroendocrine tumors), although contribute to a broader literature on the topic spanning specimens from multiple organ systems. In the current study, we did not perform a comparison of PAXgene fixation with fresh frozen tissue, which has been shown to be superior in the realm of nucleic acid preservation, but is not suitable for surgical pathology practice and, thus, was not specifically relevant to our study question. The assessment of H&E quality is inherently subjective, and preferences for staining quality vary from pathologist to pathologist.

Finally, with regard to routine molecular testing, one limitation of our study is that we did not conduct a full-scale validation of using PFPE tissues for NGS testing due to limited numbers of samples and variants in our study cohort. Although we predict comparable performance in limits of detection (LOD) based on similar coverages across detected variants, it would be essential to determine if the LOD using PFPE tissues is also comparable to FFPE tissues in a validation study assessing all classes of variants.

In summary, our results suggest that it is feasible to apply PAXgene fixative to pancreatic tissue for routine morphological diagnosis and NGS testing, with the benefit of more amplifiable gDNA. When the tissue size is small, such as being derived from ultrasound-guided fine needle aspiration [15], the greater yield of amplifiable gDNA in PFPE tissues provides important advantage over FFPE tissues.

## Figures and Tables

**Figure 1 jcm-11-04241-f001:**
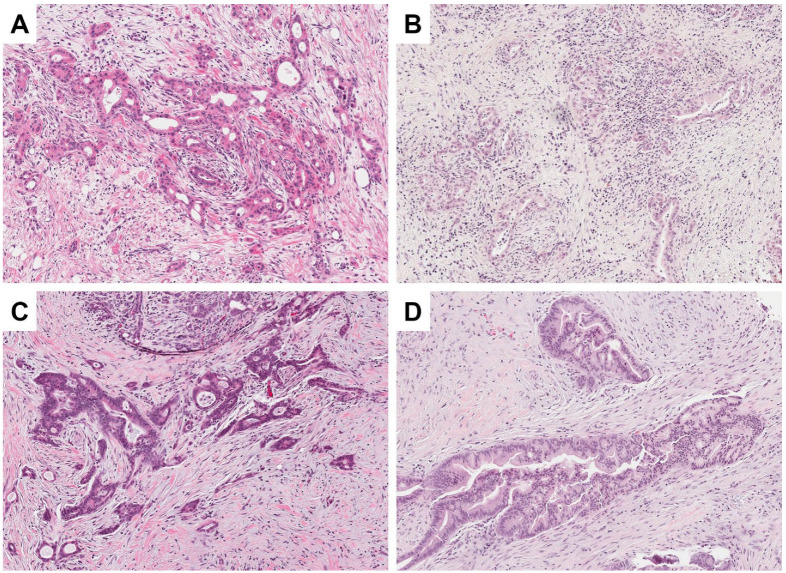
Histomorphology of PAXgene-fixed and Formalin-fixed pancreatic adenocarcinoma specimens. Most slides derived from PAXgene-fixed tissue showed good quality histomorphology with appropriate contrast ((**A**) H&E, 100×), comparable to Formalin-fixed tissue ((**C**) H&E, 100×). Occasionally, slides from both fixation methods showed a pale appearance with reduced contrast ((**B**) PAXgene, H&E, 100×; (**D**) Formalin, H&E, 100×), with no significant difference in the frequency of this finding.

**Figure 2 jcm-11-04241-f002:**
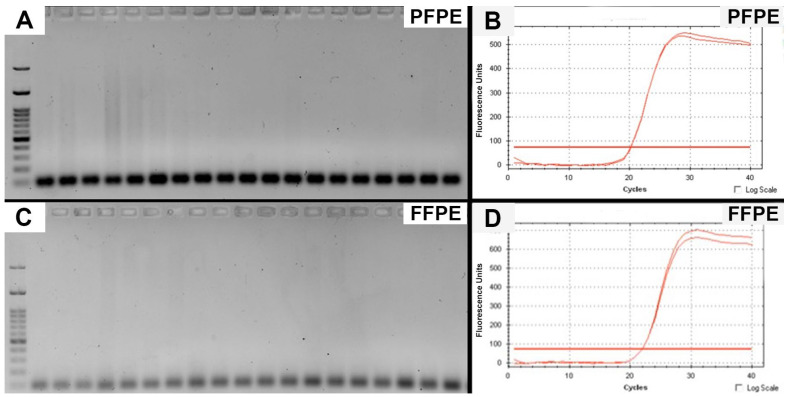
qPCR was performed on paired DNA samples in duplicate using primers for the *YWHAZ* gene. Gel electrophoresis demonstrated crisp bands at 100 bp, indicative of amplified PCR products obtained in both FFPE and PFPE samples (**A**,**C**). In all cases, PFPE samples demonstrated a significantly lower Cq value, indicative of a higher starting concentration of amplifiable DNA. Representative melt curves from one of pair samples are shown in (**B**,**D**). FFPE = formalin-fixed, paraffin-embedded; PFPE = PAXgene-fixed, paraffin-embedded.

**Table 1 jcm-11-04241-t001:** Morphological survey questions.

Do you believe the tissue on this slide is fixed with: Formalin PAXgene Independent of your answer to question 1, is the morphology on this slide adequate for diagnostic use (i.e., for routine surgical pathology signout)? Yes No If the morphology is inadequate for diagnostic use, why?

**Table 2 jcm-11-04241-t002:** Nucleic acid yields in Formalin-fixed and PAXgene-fixed tissue samples.

Result	FFPE	PFPE	P
Mean DNA concentration (ng/μL) by Qubit	36.2	44.7	0.46
Mean DNA concentration (ng/μL) by Nanodrop	118.6	144.4	0.56
Mean amplifiable DNA concentration (ng/μL) by qPCR	14.8	142.0	0.003
Mean RNA concentration (ng/μL) by Qubit	47.4	63.6	0.17
Mean RNA concentration (ng/μL) by Nanodrop	73.2	109.3	0.057
Mean amplifiable RNA concentration (ng/μL) by qPCR	5.5	7.9	0.30

Abbreviations: FFPE = formalin-fixed, paraffin-embedded; PFPE = PAXgene-fixed, paraffin-embedded.

**Table 3 jcm-11-04241-t003:** Targeted next-generation sequencing quality metrics in Formalin-fixed and PAXgene-fixed tissue samples.

Fixative	FFPE	PFPE	*p*
Median read length (bp)	111	122	0.0003
MAPD Metric	0.39	0.19	0.0007
Target base coverage at 500× (%)	85.4	97.1	0.003
Uniformity (%)	82.4	96.6	0.001
Total mapped reads	10,956,037	10,445,719	0.5
Mean depth	3052	2886	0.4

Abbreviations: bp = base pairs; FFPE = formalin-fixed, paraffin-embedded; PFPE = PAXgene-fixed, paraffin-embedded; MAPD = Median of the absolute values of all pairwise differences.

## Data Availability

Data is contained in the article, and additional data is available upon request from the corresponding author.

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
