# Peer review of "PAXgene Fixation for Pancreatic Cancer: Implications for Molecular and Surgical Pathology"

_jcm, 2022, doi:10.3390/jcm11144241_

Round 1

Reviewer 1 Report

I have reviewed the study of DeCoste R, et al. titled: “PAXgene Fixation for Pancreatic Cancer: Implications for Molecular and Surgical Pathology”. The main objective of the study was to compare the nucleic acids and histomorphologic preservation of pancreatic cancer tissues preserved with PAXgene-fixed paraffin-embedding (PFPE) and formalin-fixed paraffin-embedding. To carry out the study, 19 paired biopsies from pancreatic ductal adenocarcinoma, ampullary adenocarcinomas and two intraductal papillary mucinous neoplasms, were included. I think the work is interesting. Here are my comments:

Point 1: Methods:

Line 41-44: Following fixation of PAXgene samples, they were immersed in PAXgene Tissue Stabilizer and stored refrigerated (refrigerated at ~4oC) until processing.

In general, how long was the fixing time of the material with the new fixer? Was the size of the sample assessed to calculate the fixation time? All the tissues had the same fixation time? If there was adipose tissue in the samples, was the fixation adequate?.

Point 2: If the gold standard is fresh tissue, to obtain nucleic acids, it would be appropriate to process fresh tumor material and compare it with formalin and PAXgene fixative.

Point 3: Procedures in the pathology laboratory are standardized for formalin fixation. Histochemistry and immunohistochemistry are performed with methods for formalin-fixed material. Both histochemistry and immunohistochemistry should be performed on the study samples to assess the possibility of implementing the PAXgene fixative in daily practice.

Point 4: How many observations were required by each evaluator?

Point 5: It is convenient to include inclusion and exclusion criteria.

Results:

Line 45-46: However, there was a lack of interobserver variability with inconsistent labelling of particular slides as pale.

The results are subjective. A quantification method is necessary.

Discussion:

Line: 28-30. In our study, all H&E-stained histology slides generated from PFPE tissue were assessed as being morphologically adequate for standard diagnostic surgical pathology practice.

I believe that this assertion is not correct, since the study is limited to a particular sample and is not representative of daily practice in pathological anatomy.

What are the limitations of the study?

Author Response

Reviewer 1

Comments and Suggestions for Authors

I have reviewed the study of DeCoste R, et al. titled: “PAXgene Fixation for Pancreatic Cancer: Implications for Molecular and Surgical Pathology”. The main objective of the study was to compare the nucleic acids and histomorphologic preservation of pancreatic cancer tissues preserved with PAXgene-fixed paraffin-embedding (PFPE) and formalin-fixed paraffin-embedding. To carry out the study, 19 paired biopsies from pancreatic ductal adenocarcinoma, ampullary adenocarcinomas and two intraductal papillary mucinous neoplasms, were included. I think the work is interesting. Here are my comments:

Point 1: Methods:

Line 41-44: Following fixation of PAXgene samples, they were immersed in PAXgene Tissue Stabilizer and stored refrigerated (refrigerated at ~4oC) until processing.

In general, how long was the fixing time of the material with the new fixer? Was the size of the sample assessed to calculate the fixation time? All the tissues had the same fixation time? If there was adipose tissue in the samples, was the fixation adequate?

We thank the reviewer for raising these questions, which could help to compare our results with others in the literature, and to allow for our protocol to be repeated by others if desired. We have clarified additional details in sections 2.1 of the Methods to indicate that all of our tissue samples were approximately equal in size, falling within the maximum tissue dimensions recommended for PAXgene fixation, and did not include adipose tissue. We have also clarified that a 24 hour fixation time was utilized in all cases, followed by refrigerated storage in the tissue stabilizer for 1-4 weeks.

Point 2: If the gold standard is fresh tissue, to obtain nucleic acids, it would be appropriate to process fresh tumor material and compare it with formalin and PAXgene fixative.

We acknowledge the reviewer’s comment - fresh frozen tissue is well known to provide the best nucleic acid preservation for molecular analysis, and additionally has already been demonstrated to be superior to PAXgene fixation. Our aim with this study was to assess the utility of PAXgene fixation for routine surgical pathology and molecular pathology of pancreatic cancer specimens. Fresh frozen tissue is not suitable for surgical pathology, given the lack of morphological preservation, and so Formalin fixation is the standard fixative of choice. In that regard, we opted to analyze a direct comparison between Formalin and PAXgene. However, we have opted to acknowledge the lack of fresh frozen tissue as a limitation of our study in our revised discussion.

Point 3: Procedures in the pathology laboratory are standardized for formalin fixation. Histochemistry and immunohistochemistry are performed with methods for formalin-fixed material. Both histochemistry and immunohistochemistry should be performed on the study samples to assess the possibility of implementing the PAXgene fixative in daily practice.

The absence of histo-/immunohistochemical analyses in this study is an acknowledged limitation, which relates to resource availability for the study. We have included a paragraph acknowledging this limitation in the revised manuscript. From a practical perspective, in routine surgical pathology specimens obtained in pancreatic ductal adenocarcinoma, special stains are not typically required. Nevertheless, future study to further clarify the antigen preservation in tissues is desirable.

Point 4: How many observations were required by each evaluator? 

This point has been clarified in section 2.3 of the Methods.

Point 5: It is convenient to include inclusion and exclusion criteria.

Inclusion and exclusion criteria have now been included in the introductory paragraph of our Methods section.

Results:

Line 45-46: However, there was a lack of interobserver variability with inconsistent labelling of particular slides as pale.

The results are subjective. A quantification method is necessary.

Thank you to the reviewer for raising this issue. The sentence should have initially read “there was a lack of interobserver agreement” rather than “variability”. Nevertheless, we acknowledge these results are subjective, as the perceived adequacy of H&E staining is inherently so. Different pathologists prefer different balances of hematoxylin and eosin, different contrast, and also inherently may see colors differently. We are not aware of a purely quantitative method to assess the paleness of H&E staining. However, by accruing measures of adequacy from multiple observers and analyzing them via descriptive statistics, we have attempted to provide a semi-quantitative result with regard to the assessment of H&E quality. Notwithstanding, we have acknowledged the subjectivity of assessing histological quality in our revised discussion.

Discussion:

Line: 28-30. In our study, all H&E-stained histology slides generated from PFPE tissue were assessed as being morphologically adequate for standard diagnostic surgical pathology practice.

I believe that this assertion is not correct, since the study is limited to a particular sample and is not representative of daily practice in pathological anatomy

We thank the reviewer for raising this point. The intention of this statement was to indicate that each specimen was deemed morphologically adequate for establishing a diagnosis in a pancreatic surgical specimen, and not necessarily to imply a generalizability to other tissue types. We have altered the statement to provide a more focused assertion, and also have acknowledged the specific focus of our study in our expanded limitations (see revised discussion).

What are the limitations of the study?

We have expanded upon the limitations of our study in our revised discussion.

Reviewer 2 Report

Description of the work:

The authors verified the performance of the PAX gene tissue fixation reagent by parallel analysis of 19 prospective pancreatic cancer cases. Samples from each tissue wes fixed by formalin or PAXgene in parallel, and paraffin embedded. DNA and RNA was extracted and quality and quantity parameters measured. Amplifiability of the extracted DNA/RNA was tested by PCR. Eight samples were also subjected to NGS with the oncomine panel, based on AmpliSeq/Ion Torrent technology.

Comment:

The study focuses on a very precise aim and on a disease (pancreatic cancer) that may be relevant to many labs performing molecular analyses on the same type of samples. For this reason, the data herein presented, though interesting, should be expanded and better contextualized. Please find below a list of main points of concern.

1. PAXgene has been around for several years, the FP7 EU project that validated its use (SPIDIA), led by QIAGEN, started in 2008 and ended in 2013. Several papers (more than those cited by the authors) already reported similar results. Some focused more on IHC and ISH, other on NGS or PCR. This should be presented to the reader, clarifying what are the scope and the novelty of the present manuscript.

2. Importantly Groelz et al, researchers from QIAGEN involved in the SPIDIA project, recently (2018) published a work (https://doi.org/10.1371/journal.pone.0203608) showing that the performance of samples treated by PAXgene fix reagent and stored at RT for several years seem to degrade more than FFPE samples. This improved when PFPE/FFPE blocks were stored at 4°C or -20°C, but storage at low temperatures poses other issues that should be discussed.

3. It is true that the use of PFPE for pancreatic cancer samples was never explored, yet I believe the experiment sould be presented from a different perspective. Since pancreas is rich in nucleases, fixation with formalin is effective in removing their function because it causes the formation of covalent bonds between macromolecules. Conversely, PFPE could be less effective, because by definition is a not-crosslinking method. Indeed, its lesser stability on the long run could be partly due to this. Thus, showing that its use allows to obtain good DNA/RNA has merit. However, from the data herein presented it is impossible to know wether degradation will be superior to FFPE after some years of sample storage and this should be discussed.

4. A comparison of unit costs for FFPE vs PFPE (cost/block) should also be presented and discussed. Indeed, previous publications were hazy about the cost increment that the introduction of PAXgene in the routine processes would yield and the authors could add some really useful, yet unpublished information.

5. The authors show that PFPE samples yield longer amplicons, yet this is limited to rather short designs (around 100-150 nucleotides). Previous works showed the tracks of Agilent Tape station for isolated DNA/RNA and performed multiple PCR at increasing amplicon length (e.g. Groelz et al, Plos ONE 2018). The authors could provide similar information to show that indeed degradation is not an issue in PFPE pancreatic cancer samples, albeit when processed soon after fixation.

6. Methods: How many samples were from surgery and how many from biopsy? Were there any differences in these samples concerning amount and quality of obtained nucleic acids?

7. Methods/results: Was any IHC/ISH performed on PFPE samples? 

8. Another interesting observation regards MAPD. It is interesting that PFPE provide a stabler signal at NGS with AmpliSeq/IonTorrent technology, as high MAPD of FFPE DNA has been an issue for some time in that field. Did the authors find any difference in CNV reports from IonReporter of parallel samples extracted with PFPE vs FFPE? This would add a lot to the statement that PFPE could improve CNV detection in AmpliSeq/IonTorrent NGS.

Round 2

Reviewer 1 Report

The authors have made the recommended changes. I consider that paper could be accepted, in the present form.

Reviewer 2 Report

The authors addressed all raised issues.